# Plant Volatile, Phenylacetaldehyde Targets Breast Cancer Stem Cell by Induction of ROS and Regulation of Stat3 Signal

**DOI:** 10.3390/antiox9111119

**Published:** 2020-11-13

**Authors:** Hack Sun Choi, Su-Lim Kim, Ji-Hyang Kim, Yu-Chan Ko, Dong-Sun Lee

**Affiliations:** 1Subtropical/Tropical Organism Gene Bank, Jeju National University, Jeju 63243, Korea; choix074@jejunu.ac.kr; 2Interdisciplinary Graduate Program in Advanced Convergence Technology & Science, Jeju National University, Jeju 63243, Korea; ksl1101@naver.com (S.-L.K.); seogwi12@naver.com (J.-H.K.); koyuchan94@gmail.com (Y.-C.K.); 3School of Biomaterials Science and Technology, College of Applied Life Science, Jeju National University, Jeju 63243, Korea; 4Faculty of Biotechnology, College of Applied Life Sciences, Jeju National University, SARI, Jeju 63243, Korea; 5Practical Translational Research Center, Jeju National University, Jeju 63243, Korea

**Keywords:** cancer stem cells (CSCs), phenylacetaldehyde (PAA), breast cancer, Stat3, ROS

## Abstract

Cancer stem cells (CSCs) are undifferentiated cells that give rise to tumor and resistance to chemotherapy. This study reports that phenylacetaldehyde (PAA), a flower flavor, inhibits formation on breast CSCs. PAA showed anti-proliferation and increased apoptosis of breast cancer. PAA also reduced tumor growth in an in vivo mice model. PAA reduced the CD44^+^/CD24^−^ and ALDH1-expressing cells, mammosphere formation, and CSC marker genes. PAA preferentially induced reactive oxygen species (ROS) production and combined treatment with PAA and N-acetyl cysteine (NAC) decreased inhibition of mammosphere formation. PAA reduced phosphorylation of nuclear Stat3. PAA inhibited Stat3 signaling through de-phosphorylation of Stat3 and reduced secretory IL-6. Our results suggest that the PAA-induced ROS deregulated Stat3/IL-6 pathway and PAA may be a potential agent targeting breast cancer and CSCs.

## 1. Introduction

Breast cancer refers to all malignant tumors that occur in the breast tissue and is frequently diagnosed as solid cancer in female. Breast tumor can occur in both men and women, but it is far more common in women, covering 23% of cancer cases and 14% of cancer deaths [1]. Although mammography and chemotherapy on breast cancer has decreased the rate of death for breast cancer [2,3], this disease is still a serious disease owing to recurrence. CSCs, cancer initiating cells, comprise a small population in tumors and play an important role in cancer relapse, metastasis, and chemoresistance. First, the existence of CSCs was found in myeloid leukemia [4] and later identified in different tumors [5]. The properties of CSCs include interactions with tumor microenvironments. CSCs have a role in translational applications of cancer treatment [6].

CSCs comprise a tiny population of the tumor. The STAT, Hedgehog, Notch, Wnt, and NF-κB signaling pathways, are critical pathways of CSCs [7]. Targeting breast CSCs (BCSCs) is essential in breast cancer treatment. BCSCs express stem cell specific proteins, Oct4, Sox2, Nanog, and ALDH1 [8,9]. ALDH1 is widely known as a marker or a target of CSCs [10]. CD44^+^/CD24-expressing cancer cells have a high capacity for invasive properties [11]. Stat3 is a constitutively activated mammosphere related to the JAK/Stat3 pathway [12]. Extracellular IL-6 activates the JAK/Stat3 pathway and regulates the Oct4 gene [13]. IL-6 plays role in the transition from cancer cells into CSCs [13]. Breast CSCs can be identified by expression of biomarkers such as high CD44^high^/CD24^low^, ESA+ (Epithelial specific antigen), and ALDH1. Chemotherapy effectively increased the percentage of CD44^high^/CD24^low^-expressing cancer cells and breast CSCs formation [14,15]. CSCs induced a high level of specific ABC transporters to protect CSCs from toxins. The ABC pump is used to isolate a side population that could be sorted using ABCG2 transporter specific Hoechst 33342 dyes [16]. As BCSCs produce low levels of ROS than cancer cells, BCSCs were radio-resistance [17].

We wanted to find plant volatile organic compounds (VOCs) killing breast CSCs and started screening the VOCs against breast CSC formation using fifteen plant VOCs. We identified a CSCs inhibitor, phenylacetaldehyde (VOC) that showed inhibition of mammosphere formation. Phenylacetaldehyde (PAA) is a flower flavor that acts as an attractant for insects, and is an important rose and tomato aroma volatile. This compound is important in flavor of many foods. In this article, we focused on rose and tomato volatile organic compound (VOC), PAA derived from plants, as a candidate of CSCs inhibitor. For the first time, we show that PAA can selectively block breast CSCs formation through ROS induction/Stat3 dephosphorylation, mediated blockade of Stat3 signal pathway in breast cancer-derived mammosphere cells, and PAA is effective inhibiting tumor growth using the mouse xenograft model. These data suggested that PAA can be used in breast cancer treatment by targeting CSCs via the ROS/Stat3/IL-6 signal pathway.

## 2. Materials and Methods

### 2.1. Materials and Reagents

Tissue culture plates were purchased from Corning (Tewksbury, MA, USA). Flavors containing phenylacetaldehyde were purchased from Sigma-Aldrich Co (St. Louis, MO, USA). Chemicals such as methanol were purchased from Sigma-Aldrich Co (St. Louis, MO, USA).

### 2.2. Mammnosphere Formation Assay

Cancer cells were obtained from the American Type Culture Collection (ATCC; Manassas, VA, USA). We followed a previously described method [18] and the breast cancer cells were incubated in Dulbecco’s modified essential medium (DMEM; Hyclone, Logan, UT, USA) with 10% fetal bovine serum (FBS; Hyclone, Logan, UT, USA), 1% penicillin/streptomycin (Hyclone, Logan, UT, USA). Breast cancer cells were incubated at 37 °C in 5% CO_2_ incubator. Cancer cells were plated with 2 × 10^6^ cells in 10 cm culture dishes. For mammospheres culture, cancer cells were cultured with 3.5 × 10^4^ and 0.5 × 10^4^ cells/well with MammoCult^TM^ medium (StemCell Technologies, Vancouver, BC, Canada). The cells were incubated in a 5% CO_2_ incubator. The complete MammoCult™ medium was supplemented with 4 μg/mL heparin, 0.48 μg/mL hydrocortisone, 100 U/mL penicillin, and 100 μg/mL streptomycin. At 7 days of culture, a 6-well plate was scanned and counted using the software program NICE [19]. For mammosphere formation assay, MFE (%) was determined as previously described [20].

### 2.3. Cell Proliferation

We followed a previously reported method [18]. MDA-MB-231 and MCF-7 cells were seeded in a 96-well plate for 24 h. The breast cancer cell lines were treated with increasing concentrations (0, 0.1, 0.2, 0.5, 1, and 2 mM) using phenylacetaldehyde (100 mM PAA stock solution) for 24 h in cell culture medium. Then, we followed the cell viability assay manufacturer’s protocol CellTiter 96^®TM^ Aqueous One Solution (Promega, Madison, WI, USA) was used for cell viability assay. After mixing DMEM and aqueous one solution (5:1), we added 100 µL of mixture of each well and incubated the cells at 37 °C for 1 h. The OD_490_ was measured using a VersaMax ELISA microplate reader (Molecular Devices, San Jose, CA, USA).

### 2.4. Caspase-3/7 Assay

MDA-MB-231 cells were seeded in a 96-well plate for 24 h. The breast cancer cell was treated with increasing concentrations (0, 0.1, and 0.2 mM) using phenylacetaldehyde for 24 h in culture medium. Caspase-3/7 activity assay was performed by using the Caspase-Glo 3/7 kit (Promega, Madison, WI, USA). Then, 100 μL of Caspase-Glo3/7 reagent was added into cancer cell-cultured 96-well plates. Plates were incubated for 1 h at room temperature and were measured by using a plate-reading luminometer, GloMax^®^ Explorer (Promega, Madison, WI, USA).

### 2.5. Annexin V/PI Assay and Analysis of Cell Apoptosis

Cancer cells were cultured in 6-well plates with PAA (0.2 mM) for 24 h or DMSO. The cells were trypsinized to be single cells and washed with 1× PBS. The single cells (1 × 10^5^) were counted and suspended with 100 µL of 1× Annexin V binding buffer. We added 5 µL of FITC Annexin V solution and 5 µL of propidium iodide staining solution to each samples. Then, those were incubated for 15 min at room temperature. After washing with 1× Annexin V binding buffer, the cell pellet was analyzed using an Accuri C6 flow cytometer (BD, San Jose, CA, USA). Nuclear staining of MDA-MB-231 cells was performed by Hoechst 33342 dye. Cancer cells were treated with PAA (0.1 mM) for 24 h, and incubated with Hoechst 33342 dye (10 mg/mL) solution for 30 min at 37 °C. The dye-stained cells were observed using a fluorescence microscope (Lionheart FX, Biotek, Winooski, VT, USA).

### 2.6. Colony Formation and Migration Assay

Cancer cells were cultured in a 6-well plate with 2 × 10^3^/well and treated with PAA (0.05 and 0.1 mM) for 7 days in DMEM media. The colonies were washed three times with 1× PBS, fixed for 10 min using 4% formaldehyde, and stained for 1 h with 0.04% crystal violet. After washing twice with distilled water, we acquired images using a scanner (Epson Perfection, Epson, Tokyo, Japan). The colonies were counted with the NICE software program. The scratch was performed using a micro-tip. MDA-MB-231 cells were cultured in a 6-well plate with 2 × 10^6^ cells/plate. After the cells were washed with DMEM, cell migration was assay under PAA (0.1 mM) for 18 h. Photographs of the migrated areas were acquired using a light microscope. We followed a previously described method [21].

### 2.7. CD44^+^/CD24^−^Expression Assay

Cancer cells were cultured in a 6-well plate and treated with PAA (0.2 mM) for 1 days CD44^+^/CD24^−^Expression was assayed by FACS analysis in cancer cells. We followed a previously described method [21]. The detached cells were washed with FACS buffer and suspended with 100 µL of FACS buffer. We added 10 µL of FITC-conjugated anti-human CD44 and 10 µL of PE-conjugated anti-human CD24 to each samples. The samples were incubated on ice for 30 min, washed two times with 1X FACS buffer, and the isolated cells were then centrifuged and washed two times with 1X FACS buffer. The cell pellet was analyzed using an Accuri C6 flow cytometer (BD, San Jose, CA, USA).

### 2.8. Measurement of ROS Using DCFDA (2′, 7′-Dichlorofluorescein Diacetate)

Cancer cells were incubated in 96-well plates with PAA (0.2 mM) for 24 h. ROS were detected by using ROS-Glo™ H_2_O_2_ Detection Solution according to manufacturer’s instruction (Promega, Wisconsin, WI, USA). 100μL of ROS-Glo™ H_2_O_2_ Detection Solution was added to 96-well plate and the plate was incubated for 20 min at room temperature. Relative luminescence was measured using a plate reader, the luminometer, GloMax^®^ Explorer (Promega, Wisconsin, WI, USA). PAA-treated cells were incubated with 10 μM DCFDA dye for 20 min. DCFDA dye was removed and washed with 1XPBS. Fluorescent cell was observed under a fluorescent microscope (Lionheart FX).

### 2.9. Real-Time RT-qPCR

Total RNA was isolated using MDA-MB-231 cells. RT-qPCR was performed using One-Step RT-qPCR kit (Enzynomics, Daejeon, Korea). We followed a previously described method [18]. We made the RT-qPCR mixture containing TOPreal^TM^ One-Step RT qPCR Enzyme MIX 1 µL, 2XTOPreal^TM^ One-Step RT qPCR Reaction MIX (with low ROX) 10 µL, RNA template (100 ng/µL) 1 µL, specific primers-F (10 ng/µL) 1 µL, specific primers-R (10 ng/µL) 1 µL, and sterile water 6 µL in each sample. Specific primers can be found in Table 1. The β-actin gene was used as a control for RT-qPCR.

### 2.10. ALDEFLUOR Assay

The ALDH activity of MDA-MB-231 cells was examined using an ALDEFUOR^TM^ assay kit (STEMCELL Technologies, Vancouver, BC, Canada). We used a previously described method [21]. The active reagent BODIPY-aminoacetaldehyde was added to breast cancer cells and was converted to fluorescent BODIPY-aminoacetate by ALDH. The ALDH inhibitor diethylaminobenzaldehyde (DEAB) was used as a negative control. The cancer cells were treated with PAA (200 µM) for 24 h and incubated in ALDH assay buffer at 37 °C for 30 min. The ALDH-positive cells were assayed with an Accuri C6 (BD, San Jose, CA, USA).

### 2.11. Immunoblot Blotting

Proteins isolated from breast cancer and mammospheres treated with PAA (0.2 mM) for 24 h were separated by using a 10% SDS-PAGE gel and SDS-PAGE was conducted using tris-glycine buffer. After transferred, the polyvinylidene fluoride (PVDF) membranes (Millipore, Burlington, MA, USA) were incubated with Odyssey blocking buffer at room temperature for 1 h and then incubated overnight with primary antibodies at 4 °C. The primary antibodies were diluted one thousand to one with primary antibody diluent (mixture of Odyssey blocking buffer and 0.2% tween-20). The primary antibody was phospho-Stat3 (Cell Signaling Technology, Danvers, MA, USA), and Stat3, p65, Lamin B, and β-actin (Santa Cruz Biotechnology, Dallas, TX, USA). After the PVDF membranes were washed three times using PBS-Tween 20 (0.1%, *v*/*v*), all membranes were incubated with IRDye 680RD and IRDye 800W-labeled secondary antibodies with secondary antibody diluent (mixture of Odyssey blocking buffer, 0.2% tween-20, and 0.01% SDS) for 1 h at room temperature. The signals were detected with an Odyssey CLx imaging system (LI-COR, Lincoln, NE, USA) at the Jeju Center of Korea Basic Science Institute (KBSI, core-facility center).

### 2.12. Electrophoretic Mobility Shift Assays (EMSA)

Nuclear proteins were prepared as described previously [22]. EMSAs were performed by using a Lightshift chemiluminescent EMSA kit according manufacturer’s instruction (Thermoscientific, IL, USA). The biotin-upper and biotin-lower strand of Stat3 probe (5′-cttcatttcccggaaatcccta-Biotin-3′ and 5′-tagggatttccgggaaatgaag-Biotin3′) were used. The biotin-labeled DNA probes were incubated with PAA-treated nuclear proteins in a final volume of 20 μL EMSA buffer containing 1 μg/μL poly (dI-dC) at room temperature for 20 min. The protein/DNA complexes were run on a 4% polyacrylamide nondenaturing gel in 0.5× TBE (45 mM tris-borate and 1 mM EDTA) and detected using a chemiluminescent nucleic acid detection kit (Thermo Scientific).

### 2.13. Small Interfering RNA (siRNA)

To determine the effect of Stat3 on mammosphere formation, we treated MDA-MB-231-derived mammospheres with human Stat3 siRNA (Bioneer, Daejeon, Korea). The following siRNAs were used: Stat3 siRNA (NM_ 1145658). The scrambled siRNA control was a nontargeting siRNA from Bioneer. For transfection of siRNA, MDA-MB-231 cells were seeded into 6-well plates for 24 h, and then, transfection of siRNA was performed using Lipofectamine 3000 (Thermo Scientific) according to the manufacturer’s protocol. The Stat3 protein level was determined by Western blot using anti-Stat3.

### 2.14. Tumor Xenografts

Female four-week-old nude mice obtained from OrientBio (Seoul, Korea) were kept in independent-ventilation cages with access to food and water for one week. Twelve BALB/C female nude mice were injected with MDA-MB-231 cells (2 × 10^6^ cells/mouse) and PAA (10 mg/kg = 200 μg/mouse) was intraperitoneally administered on one time/week for 4 weeks. Mouse tumor volumes were measured and calculated for one month using the formula (width^2^ × length)/2. Animal care and experiments were performed in accordance with protocols approved by the Institutional Animal Care and Use Committee (IACUC) of Jeju National University (JNU-IACUC; Approval Number 2018-003). The mouse experiments were performed as described previously [23]. Tumor volumes were measured using the formula, V = (width^2^ × length)/2.

### 2.15. Statistical Analysis

All data are presented as the mean ± standard deviation (SD). Data were analyzed using Student’s *t*-test. A *p*-value lower than 0.05 was considered statistically significant (GraphPad Prism 5 Software).

## 3. Results

### 3.1. Screening of the Effect of Volatile Organic Compounds (VOCs) Derived from Plants on Mammosphere Formation

The aim at this study is to isolate volatile organic compounds (VOCs) from plants for killing cancer stem cells derived from breast cancer. To screen the VOCs against breast CSC formation, we screened representing VOCs consisting of 15 plant flavors by mammosphere counting [20]. We identified a VOC inhibitor that showed a more than 50% inhibition of mammosphere formation efficiency (MEF) (Table 2). Plant volatile flavor, PAA (Figure 1A) was chosen as a strong inhibitor of BCSC formation.

### 3.2. PAA Inhibits Cell Proliferation and Increased Apoptosis of Breast Cancer

We tested proliferation on breast cancers using PAA. PAA has an anti- proliferation effect on breast cancers (Figure 1B,C). PAA inhibited the proliferation of MDA-MB-231 and MCF-7 cells in a dose-dependent manner. The doses of PAA causing 50% growth inhibition (IC_50_) of MDA-MB-231 and MCF-7 cells at 24 h incubations were 550 and 606 µM. The number of apoptotic (Annexin V+) breast cancer cells were increased by PAA at 200 μM (Figure 1D). The subpopulation of cells in early apoptosis increased from 14.5% to 22.7%. A Caspase-3/7 activity showed that PAA increase caspase activity (Figure 1E). The apoptotic bodies of breast cancer cells were induced under PAA (Figure 1F). The PAA blocked migration and colony formation of breast cancer cells (Figure 1G,H). Our data indicated that PAA effectively inhibits proliferation, migration, and colony formation and induced apoptosis.

### 3.3. PAA Inhibits Tumor Growth in In Vivo

As PAA inhibits cell proliferation in vitro, we test whether PAA inhibited tumor growth in in vivo. Tumor volume of PAA-treated mice was smaller than the control mice (Figure 2A,B). Tumor weights of PAA-treated mice were lower than the control mice (Figure 2C). PAA-treated mice showed similar body weights as the control mice group (Figure 2A). Our results showed PAA inhibits tumor growth in the in vivo mouse model.

### 3.4. PAA Inhibits Breast CSCs

To examine whether PAA can inhibit mammosphere formation, we treated PAA with the primary mammosphere derived from breast cancer cells. At Figure 3, PAA reduced the mammospheres formation. Numbers of mammosphere were declined by 50% to 90% and the size of mammospheres was reduced (Figure 3A,B). The most important flavors in rose and tomato, PAA and phenylethanol are derived from phenylalanine (Figure 3C) [24]. To evaluate whether phenylalanine-derived flavors, PAA, phenylethanol, and structural analogues (2- and 3-phenylpropioinaldehyde), can inhibit the formation of tumorspheres, we added these volatiles at the mammosphere. In Figure 3D, PAA and 2-phenylpropioinaldehyde strongly inhibited mammosphere formation. Furthermore, 3-phenylpropionaldehyde and phenylethanol inhibited mammosphere formation.

### 3.5. The PAA Reduced CD44^+^/CD24^−^-Expressing Population and ALDH-Positive Cells

Cancer cells were incubated with PAA (0.2 mM) for 1 day and CD44^+^/CD24^−^-expressing population on breast cancer was tested. PAA decreased CD44^+^/CD24^−^-expressing subpopulation of cancer cells from 47.5% to 21.3% (Figure 4A). Cancer cells were incubated with PAA (0.2 mM) for 1 day and used ALDEFLUOR assay kit to test effect of PAA on ALDH-expression cells. PAA reduced ALDH-positive cells from 1.1% to 0.2% (Figure 4B). PAA inhibits breast CSCs hallmarks, CD44^+^/CD24^−^-expressing and ALDH-positive cells.

### 3.6. PAA Inhibits CSC-Specific Gene Expression and Proliferation of Mammospheres

To examine whether PAA regulates CSC-specific gene expression, we examined transcription of CSC-specific genes using RT-qPCR. PAA reduced Nanog, Sox2, Oct4, and CD44 gene transcripts of cancer cells (Figure 5A). To confirm that PAA inhibits proliferation of mammosphere, we incubated PAA on mammospheres and counted cell numbers. PAA reduced cell number of PAA-incubated mammospheres and killed cells of mammospheres. These data indicated that PAA lead to a dramatic reduction of mammosphere growth (Figure 5B).

### 3.7. PAA Induce ROS and N-Acetycysteine (NAC) Reversed PAA-Induced Inhibition of Mammosphere Formation

Induced ROS kills CSCs and low levels of ROS induced formation of CSC [25]. We examined the ROS level on BCSCs under PAA. To test ROS production induced by PAA, we performed DCF-DA staining. In Figure 6A,B, PAA increased the ROS level in breast cancer cells. To test PAA-dependent ROS production, we treated N-acetylcysteine on CSCs culture. NAC pretreatment attenuated the inhibition of PAA-induced mammosphere formation. Our data indicated that PAA-induced ROS production increases CSC death (Figure 6C).

### 3.8. Effect of PAA on Stat3 Signal Pathway Regulation through ROS Production

Because Stat3 is survival factor of CSCs, we check Stat3 signaling to BCSC formation of 0.2 mM PAA. PAA treatment reduced the phosphorylation level of Stat3 in BCSCs (Figure 7A) and NAC reversed PAA-induced CSCs inhibition (Figure 7B). In addition, after treatment with PAA, we examined the phosphorylation of nuclear pStast3. The phosphorylation of nuclear pStat3 was reduced under PAA (Figure 7C). Together, these results suggested that PAA downregulated the Stat3 signaling pathway through ROS production.

### 3.9. The PAA Inhibits Stat3 Signal-Dependent IL-6 Production on Mammospheres

To find targets of PAA, we examined Stat3 signal in the mammosphere under treatment of PAA. PAA could decrease the nuclear pStat3 level compared to the control and could not decrease the nuclear p65 level (Figure 7C). We checked Stat3/DNA binding to PAA-treated nuclear proteins using EMSA experiments. In Figure 7D, PAA reduced the binding level of Stat3 protein/Stat probe (Figure 7D, lane 3). Stat3 protein/Stat probe binding was confirmed using 10× competitor oligo (Figure 7D, lane 4) and 10× mutated-Stat oligo (Figure 7D, lane 5). PAA reduced Stat3/DNA-binding capacities. Extracellular IL-6 is essential for BCSCs formation [26]. pStat3 bound to the promoter region of IL-6 and regulated IL-6 transcripts. To examine extracellular IL-6 levels, we tested immuno blot analysis using mammosphere broth using anti-IL-6 antibody. Under PAA, immuno blot showed that PAA reduced the protein level of extracellular IL-6. The numbers of mammospheres with/without PAA were quantified as an internal control (Figure 7E). Our data suggested that the Stat3 signal regulates mammosphere growth and that PAA inhibits mammosphere formation by the Stat3 signaling pathway.

### 3.10. Effect of PAA Vapor on Breast CSCs Formation and pStat3 Level

To examine the effect of the PAA vapor on mammosphere formation, we analyzed CSC formation through the vaporization of PAA. The vapor PAA inhibited CSC formation (Figure 8A). In order to examine pStat3 levels on the mammosphere using PAA vapor. PAA vapor reduced pStat3 levels of the mammosphere (Figure 8B). To confirm the function of Stat3 as a CSC survival factor, we determined the effect of siRNA-induced silencing of Stat3. SiRNA of Stat3 reduced the formation of mammospheres in MDA-MB-231 cells (Figure 8C).

## 4. Discussion

The aim at this study is to isolate volatile organic compounds (VOCs) from plants for killing cancer stem cells derived from breast cancer cells. A number of volatile organic compounds represent carcinogen and have caused cancer. A few of these volatile organic compounds, formaldehyde, naphthalene para-dichlorobenzene, chloroform, acetaldehyde, and benzene, are considered to be probable human carcinogens by many authorities. Volatile organic compounds from tobacco smoke are associated with cancer. Electronic cigarettes-produced VOCs such as acrylamide, benzene, and propylene oxide, may pose health risks to nonsmoking users [27]. Plant VOCs are chemical substances produced by plants in gaseous form. Plant volatiles have a role in providing direct defense against biotic and abiotic stressors. Plant volatiles transmit signaling within and between plants [28]. Some papers showed that volatile compounds from leaves and fruits have a novel source of anticancer drugs [29,30,31]. We wanted to find plant VOCs killing breast cancer stem cells and start screening the VOCs against breast CSC formation using 15 plant VOCs. We identified a CSCs inhibitor (VOC) that showed more than 50% inhibition of mammosphere formation efficiency (MFE) (Table 2). Plant volatile flavor PAA (Figure 1A) was chosen as a strong inhibitor of BCSC formation.

Original CSCs research is based on leukemia stem cells. As CSCs of primary cancer tissue have a limited supply, CSCs of cancer cell have been chosen for CSC study. PAA as an anti-tumor and anti-CSCs shows the below: (1) PAA had anti-proliferation and induced cell apoptosis (Figure 1); (2) PAA reduced cell migration and colony formation of breast cancer (Figure 1); (3) PAA inhibits in vivo tumor growth (Figure 2); (4) PAA reduced size and formation of the mammosphere (Figure 3). PAA reduced CD44^+^/CD24^−^ and ALDH-expression cells (Figure 4); (5) PAA induced ROS production and decrease of mammosphere formation (Figure 6). (6) PAA inhibits the mammosphere formation by blocking Stat3 signal. PAA reduced extracellular IL-6 level, which is an essential cytokine of CSC (Figure 7) [32] and lead to reduction in mammosphere growth (Figure 5). PAA, a plant flavor compound, can be a new anti-cancer compound for cancer treatment by targeting CSCs.

As the most important tomato and rose aroma flavor, PAA, is derived from phenylalanine and is a major contributor to scent in many flowers. The biological function of PAA is defense and reproduction [24]. PAA was formed from amino acid, phenylalanine through the pyridoxal-5′-phosphate-dependent aromatic amino acid decarboxylase enzyme. Rose flowers and tomato fruits produce a flavor compound, PAA. Due to the importance of the rose industry, the rose scent is an attractive target for the perfume industry. PAA has important biological functions in plants and is a potent insect attractant. The presence of PAA is the attractiveness to mammals and seed dispersers [33]. In this study, we found that for the first time, PAA, a natural flavor of phenylalanine, kills breast cancer and CSCs.

Redox regulation of CSC is regarded as a good target for tumor therapy, as an important factor for the maintenance of CSC are lower levels of ROS. Prx2, an antioxidant enzyme, is important for maintaining stemness by redox regulation in liver cancer cells [34]. Our results showed that we hypothesized that the ROS regulating system of CSCs is poorer and the Stat3 signal can be considered a target of CSCs chemotherapy. We showed that PAA induces dephosphorylation of Stat3 and then blocks the Stat3 signal pathway. Recent studies have showed that metformin, an antidiabetic medication targeting CSCs in solid tumor, inhibits inflammatory response-related CSC growth and preferentially inhibits phosphorylation of Stat3 in breast CSCs [35].

Stat3 signaling is essential for BCSC formation. The nuclear translocation of NF-κB and pStat3 by PAA was tested in the mammosphere of breast cancer. PAA decreased the nuclear pStat3 (Figure 7C). However, PAA could not reduce nuclear p65 and specifically regulate the NF-κB pathway. PAA inhibits the Stat3 signal pathway through reduction of nuclear pStat3 and Stat3 /DNA binding (Figure 7D). Tumors are heterogeneous and regulate dynamic reaction between BCSCs and cancer cells by extracellular IL-6 and IL-8 [32]. A extracellar IL-6 and IL-8 regulated self-renewal of CSCs [36]. PAA inhibits the extracellular IL-6 and may be a blockade of extracellular IL-6 (Figure 7E) because high levels of extracellular IL-6 are related with a poor prognosis in cancer patient [37].

PAA can target BCSCs through mammosphere formation, Aldelfluor, and mammosphere growth assay. We suggested a mechanism by which PAA targeted BCSCs by inhibition of the ROS-induced Stat3 signal and extracellular IL-6. Our data showed the possible usage of volatile plant flavor PAA as cancer chemoprevention.

## 5. Conclusions

Phenylacetaldehyde (PAA), a flower flavor, inhibits formation on BCSCs. PAA reduced cell proliferation and increased apoptosis of breast cancer. PAA also inhibited tumor growth in the in vivo mouse. PAA decreased the CD44^+^/CD24^−^ and ALDH1-expressing cell, mammosphere formation, and CSC marker genes. PAA preferentially induced reactive oxygen species (ROS) production and induced the decrease of Stat3 phosphorylation. Combined treatment with PAA and N-acetyl cysteine (NAC) reduced Stat3 phosphorylation. Therefore, PAA inhibits CSCs formation through the ROS-dependent Stat3/IL-6 signal. PAA inhibited Stat3 signaling through dephosphorylation of Stat3 and reduced secretory IL-6. Our data suggest that PAA is useful for breast cancer treatment and the Stat3/IL-6 signal as a marker for CSCs targeting.

## Figures and Tables

**Figure 1 antioxidants-09-01119-f001:**
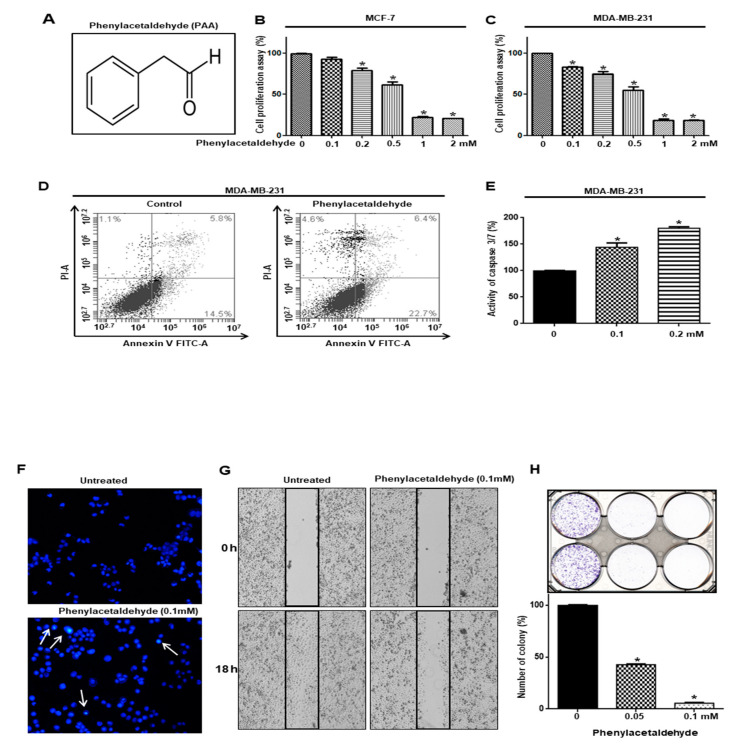
PAA inhibits proliferation, migration, and colony formation and induces apoptosis of breast cancer. (**A**–**C**) Molecular structure of PAA and effect of PAA on proliferation breast cancer cells. Breast cancer cells were treated with PAA. The anti-proliferation effect of PAA was assay by MTS assays. (**D**) The 0.2 mM PAA induced apoptosis of cancer cells. PAA-induced apoptotic cells were analyzed by using Annexin V-PI staining kit. (**E**) The caspase3/7 activity of cancer cells was determined by the Caspase-Gloss 3/7 kit. (**F**) Analysis of apoptotic cells by fluorescence staining. Nuclei from breast cancer were stained with Hoechst 33258 (magnification, ×100). The arrows indicate apoptotic bodies. (**G**) The effects of PAA on the migration of breast cancer cell line. Migrations with/without PAA were imaged at 0 and 18 h. PAA has inhibitory function of migration on cancer cells. PAA-treated wound healing area was captured. (**H**) PAA has inhibitory function of colony formation on cancer cells. The dissociated cancer cells were cultured in plates and treated with PAA. Representative images were captured. The data represent the mean ± SD of three independent experiments. * *p* < 0.05 vs. control.

**Figure 2 antioxidants-09-01119-f002:**
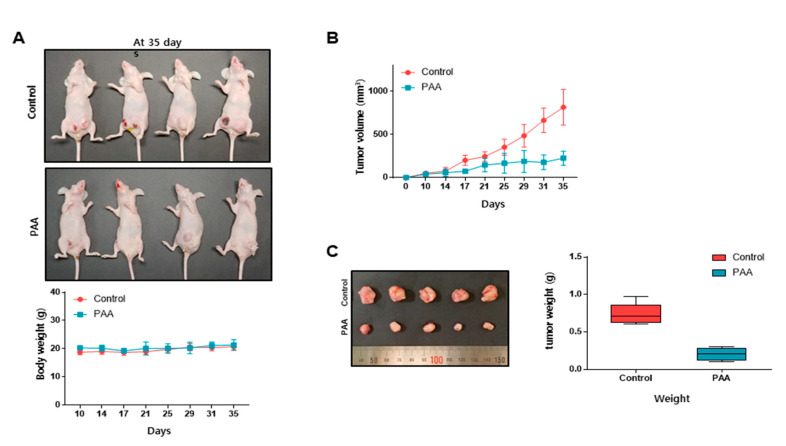
PAA inhibits tumor growth in the in vivo mice model. Cancer cells were injected into the mammary fat pad on female nude mice. (**A**) The effect of tumor growth with PAA in the female nude mice containing breast cancers. The dose used was 10 mg/kg. After 5 weeks, images were captured. (**B**) Tumor volume was measured at indicated time points and calculated as (width^2^ × length)/2. (**C**) Effect of PAA on tumor weights. Tumor weights were assayed after sacrifice. Photography of isolated tumors from the control or PAA-treated mice.

**Figure 3 antioxidants-09-01119-f003:**
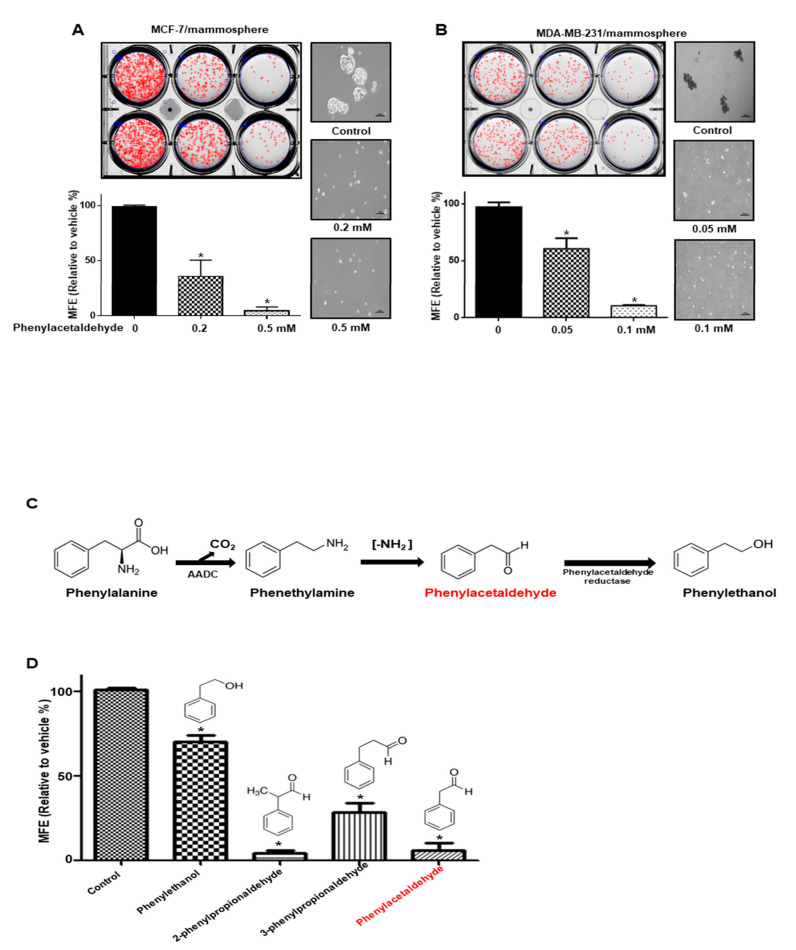
The inhibitory effect of PAA on mammosphere formation. The breast cells were incubated in mammosphere-forming media. (**A**) Inhibitory effect of PAA on mammosphere formation. Primary mammosphere from MCF-7 were incubated with PAA (0.2 and 0.5 mM) or DMSO. (**B**) Inhibitory effect of PAA on formation of mammosphere from breast cancer cells. Mammospheres were cultured with PAA (0.05 and 0.1 mM). (**C**) The synthetic pathway for flavor compounds PAA and phenylethanol derived from rose and tomato. (**D**) Effect of PAA and related volatile compounds (100 μM) (phenylethanol, 2-phenylpropionaldehyde and 3-phenylpropionaldehyde) on mammosphere formation derived from breast cancer. The data represent the mean ± SD of three independent experiments. * *p* < 0.05 vs. control.

**Figure 4 antioxidants-09-01119-f004:**
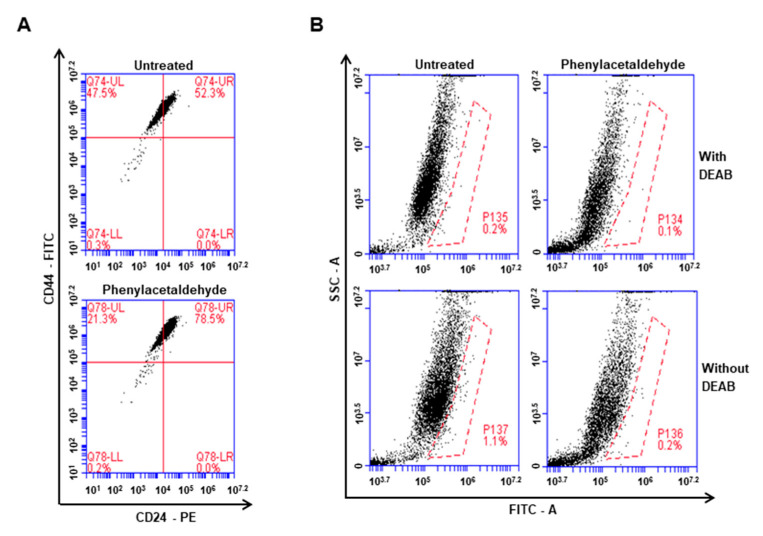
Effect of PAA in breast CD44^+^/CD24^−^ and ALDH1-expressing cell. CD44^+^/CD24^−^-cell population was assayed by flow cytometer with a PAA (0.2 mM). We followed a previously described method [21]. (**A**) 1 × 10^6^ cells were incubated with FITC-CD44 and PE-CD24 (BD, San Diego, CA, USA) and put on ice for 20 min. Breast cancers were washed two times with 1× PBS and assayed using flow cytometer (Accuri C6). (**B**) Breast cancer cells were treated with PAA (0.2 mM) for 1 day. ALDH assay was performed using ALDEFUOR kit (StemCell Technologies). We followed a previously described method [21]. Breast cancer cells were incubated in ALDH assay buffer at 37 °C for 20 min. ALDH-positive cells were determined by using personal flow cytometer (Accuri C6).

**Figure 5 antioxidants-09-01119-f005:**
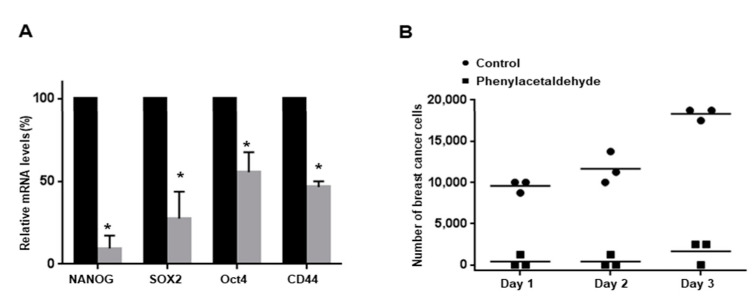
Effect of PAA on CSC-specific gene expression and proliferation of mammospheres. (**A**) The Nanog, Sox2, Oct4, and CD44 gene transcripts were determined using total RNA from PAA-treated mammospheres. (**B**) The inhibitory effect of PAA on the formation of mammospheres. PAA prevented mammosphere formation. The PAA-treated mammospheres were plated in 6-cm dishes with equal numbers of cells. One day after plating, the cells were counted. At 2 and 3 days after seeding, the cells were counted. * *p* < 0.05 vs. control.

**Figure 6 antioxidants-09-01119-f006:**
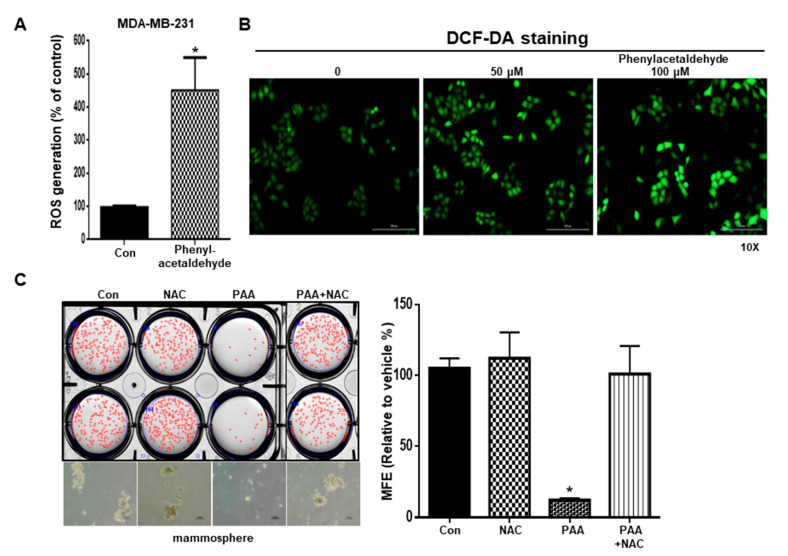
Effect of PAA-induced ROS induction and N-acetylcysteine (NAC) co-treated PAA on mammosphere formation. PAA-induced ROS generation was examined using ROS-Glo™ and DCF-DA detection assays using fluorescent microscopy. Mammospheres were pretreated with/without NAC (10 mM) 30 min prior to treatment with 0.1 mM PAA. Following culture for 7 days, mammosphere formation was examined (**A**–**C**). The mammospheres were cultured under PAA, NAC, and PAA+NAC. The data represent the mean ± SD of three independent experiments. * *p* < 0.05 vs. control.

**Figure 7 antioxidants-09-01119-f007:**
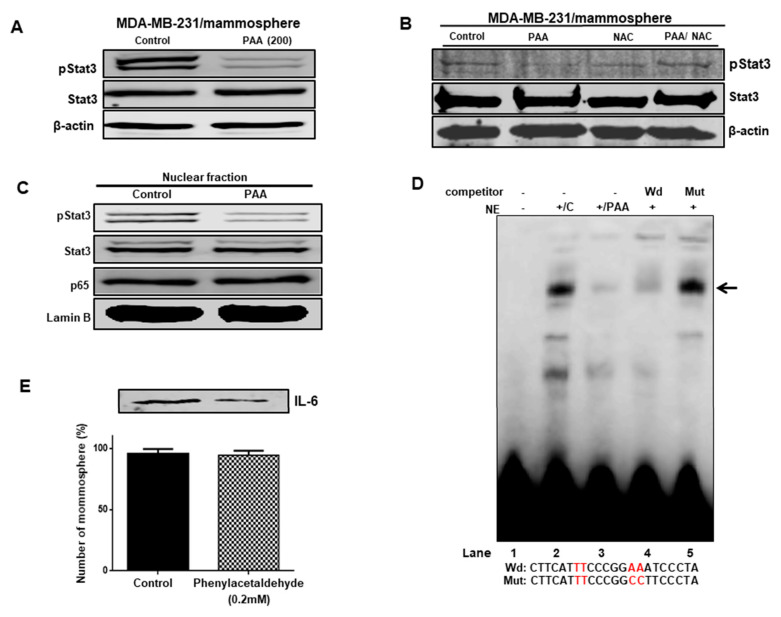
Effect of PAA on Stat3 signal and level of extracellular IL-6 on mammospheres derived from breast cancer cell. (**A**,**C**) Total and nuclear pStat3 and nuclear NF-kB were determined in mammosphere under PAA using anti-pStat3, ant-Stat3, anti-p65, anti-Lamin B, and anti-β-actin. PAA reduced nuclear pStat3 protein in mammospheres. (**B**) Effect of PAA and NAC co-treated PAA on pStat3 level of breast cancer mamospheres. PAA induced decrease of pStat3 levels of mammospheres and NAC co-treated PAA recovered phosphorylation level of PAA-induced pStat3. (**D**) EMSA of nuclear lysates treated with PAA. Nuclear proteins were incubated with biotin-labeled Stat3 oligo and run by 4 % native PAGE. Lane 1: probe only; lane 2: nuclear proteins with probe; lane3: PAA-treated nuclear proteins with probe; lane 4: 10× self-competitor oligo; lane 5: 10× mutated Stat3 oligo. The PAA decreased DNA/Stat3 binding of nuclear proteins. (**E**) Immunoblot analysis of extracellular broth from mammosphere cultures using anti-IL-6 and the number of mammospheres as controls. PAA decreased the level of extracellular IL-6 in the mammospheres.

**Figure 8 antioxidants-09-01119-f008:**
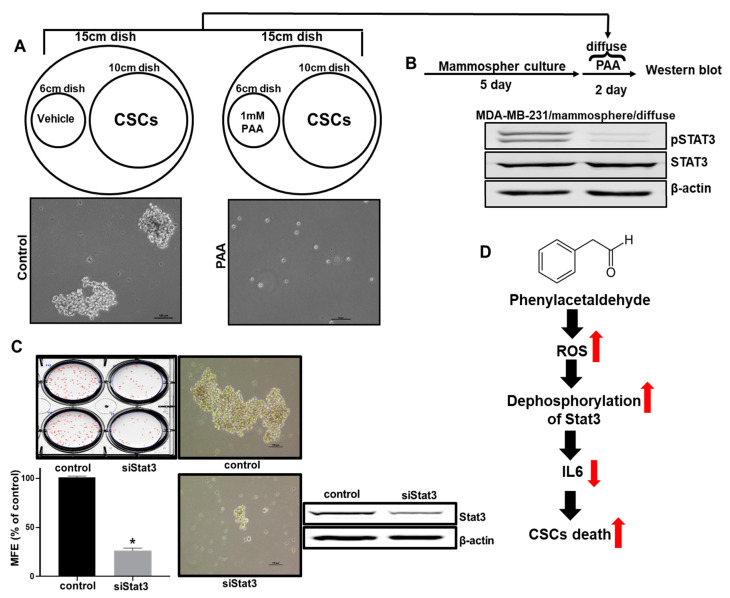
The inhibitory effect of vapor PAA on mammosphere formation. The breast cells were incubated in mammosphere-forming media under vapor PAA (1 mM methanol-dissolved PAA). (**A**) The effect of PAA vapor on mammosphere formation. Cancer cells were incubated under PAA vapor (1mM) and vehicle (methanol) using CSC culture media. Mammosphere-cultured images were obtained by microscopy. Vapor PAA killed the mammosphere. (**B**) The effect of vapor PAA on pStat3 level. Five day-cultured masmmospheres were incubated with vapor PAA for 2 days. Levels of pStat3 under vapor PAA were examined using anti-pStat3 and anti-Stat3. (**C**) The effect of Stat3 protein on mammosphere formation using siRNA of Stat3. The data shown represent the mean ± SD of three independent experiments. * *p* < 0.05 vs. DMSO-treated control. (**D**) The proposed model for CSC death by PAA. PAA induced ROS production and downregulated pStat3 levels. The PAA-induced ROS decreased phosphorylation of Stat3 and decreased the level of extracellular IL-6. Extracellular IL-6 can convert cancer cells to CSCs. PAA inhibits conversion from cancer cells to CSCs through de-phosphorylation of Stat3 through ROS production.

**Table 1 antioxidants-09-01119-t001:** Specific primer sequences of real-time RT-qPCR.

Genes	Accession Number	Primers
CD44	NP_0006013	Forward: 5′-AGAAGGTGTGGGCAGAAGAA-3′ Reverse: 5′-AAATGCACCATTTCCTGAGA-3′
Nanog	NP_0248654	Forward: 5′-ATGCCTCACACGGAGACTGT-3′ Reverse: 5′-AAGTGGGTTGTTTGCCTTTG-3′
Sox2	NP_0030971	Forward: 5′-TTGCTGCCTCTTTAAGACTAGGA-3′ Reverse: 5′-CTGGGGCTCAAACTTCTCTC-3′
Oct4	NP_1887022	Forward: 5′-AGCAAAACCCGGAGGAGT-3′-3′ Reverse: 5′-CCACATCGGCCTGTGTATATC-3′
β-actin	NP_0010921	Forward: 5′-TGTTACCAACTGGGACGACA-3′ Reverse: 5′-GGGGTGTTGAAGGTCTCAAA-3′

**Table 2 antioxidants-09-01119-t002:** The effect of volatile organic compounds derived from plants on breast cancer stem cells formation.

Number	Name	Molecular Weight	Final Concentration	Inhibitory Activity	Source (Plant)
1	Methyl salicylate	152.15	100 μM	(−)	Wintergreen
2	Methyl jasmonate	224.30	100 μM	(−)	Jasminum
3	Acetyl salicylic acid	180.16	100 μM	(−)	Meadowsweet
4	Methyl cinnamic acid	162.19	100 μM	(−)	Strawberry
5	Methyl trans-cinnamic acid	162.19	100 μM	(−)	Strawberry
6	2-Methyl cinnamic acid	162.19	100 μM	(−)	Strawberry
7	Trans-cinnamic acid	148.16	100 μM	(−)	Strawberry
8	Cis-3-hexen-1-ol	100.16	100 μM	(−)	Strawberry
9	Trans-2-hexenal	98.14	100 μM	(−)	Tomato
10	phenylacetaldehyde	120.15	100 μM	(+)	Rose
11	Trans-2-hexenol	100.16	100 μM	(−)	Banana
12	1-octen-3-ol	128.21	100 μM	(−)	Lemon balm
13	(+) β-citronellol	156.27	100 μM	(−)	Cymbopogon
14	(+) Rose oxide	154.25	100 μM	(−)	Rose
15	hexanal	100.16	100 μM	(−)	Strawberry

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
