# Peer review of "Plant Volatile, Phenylacetaldehyde Targets Breast Cancer Stem Cell by Induction of ROS and Regulation of Stat3 Signal"

_antioxidants, 2020, doi:10.3390/antiox9111119_

Round 1

Reviewer 1 Report

In this article the authors evaluated the effect of phenylacetaldehyde (PAA). In particular they show that PAA inhibieted Stat-3 signaling and also reduced the cell migration and mammosphere formation.

In many experiments it is not evident the time and the concentration of PAA that was used.

The authors should specify how PAA treatment was conducted in MDA-MB231 cells and in xenograft

Figure 8: indicate more clearly the tests loaded in the western blotting experiment;

the authors should inhibit or by using a sirna or an inhibitor of Jak activity, STAT phosphorylation to validate their hypothesis.

Author Response

We submit the reviewer 1' comment.

Reviewer 2 Report

In this study the authors have analyzed the volatile organic compound phenylacetaldehyde (PAA), a flower flavor, as a candidate inhibitor of breast cancer stem cells (CSCs) formation.

To my view, the strength of this paper could be the novelty since there is no or very limited literature in this topic and the overall findings could be of interest for the research field. On the other hand, the main weakness is the low-quality of the manuscript, which should be completely rewritten, especially because of the language used. Indeed, I found its reading very difficult. Besides, many parts need to be improved, as many details of the whole experimental design are lacking.

Firstly, the aim of the study should be clearly defined and additional information and literature data should be provided about the role exerted by these volatile organic compounds, isolated from plants, on carcinogenesis. Besides, the choice of studying their effects on CSCs should be outlined.

Moreover, all the experimental procedures used in their investigation need to be accurately detailed. For instance, MTS is mentioned in the legend of Figure 1, but the method used to assess cell proliferation is nebulous. Also, colony formation and migration assays should be specified better in the method section. This concern is spread along the whole text, mainly in the description of methods and results.

The authors have screened a series of compounds, which are described in table 1; their source should be included in the table. Furthermore, to avoid confusion, this table should precede Figure 1.

Why did the authors evaluate the concentration of 100 micromolar in this screening? Have they any knowledge on the effects of these compounds on normal cells?

Author Response

We submit reviewer'2 comments.

Round 2

Reviewer 1 Report

The authors replied to my observations. The manuscript has improved over the previous version

Author Response

The reviewer’s point is well taken and did English editing.

We attached the Certificate of English Editing.

Reviewer 2 Report

The authors have responded to all the raised comments. However, the text still needs an extensive language editing.

Author Response

(The authors gave the same response as above.)
